# The Optical and Thermo-Optical Properties of Non-Stoichiometric Silicon Nitride Layers Obtained by the PECVD Method with Varying Levels of Nitrogen Content

**DOI:** 10.3390/ma15062260

**Published:** 2022-03-18

**Authors:** Stanisława Kluska, Maria Jurzecka-Szymacha, Natalia Nosidlak, Piotr Dulian, Janusz Jaglarz

**Affiliations:** 1Faculty of Materials Science and Ceramics, AGH University of Science and Technology, Mickiewicza Avenue 30, 30-059 Krakow, Poland; kluska@agh.edu.pl; 2Department of Physics, Cracow University of Technology, Podchorążych Street 1, 30-084 Krakow, Poland; nnosidlak@pk.edu.pl; 3Faculty of Chemical Engineering and Technology, Cracow University of Technology, Warszawska Street 24, 31-155 Krakow, Poland; piotr.dulian@pk.edu.pl; 4Institute of Materials Engineering, Cracow University of Technology, Jana Pawła II Avenue 37, 31-864 Krakow, Poland

**Keywords:** PECVD technique, spectroscopic ellipsometry, amorphous SiN_x_:H layers, thermo-optical properties

## Abstract

In this paper, we investigated the optical and thermo-optical properties of a-SiN_x_:H layers obtained using the PECVD technique. SiN_x_:H layers with different refractive indices were obtained from silane and ammonia as precursor gases. Surface morphology and chemical composition studies were investigated using atomic force microscopy, scanning electron microscopy, Fourier transform infrared spectroscopy and energy dispersive spectrometry methods. Spectroscopic ellipsometry was used to determine the optical indexes, thicknesses and optical bandgap of the films. The main purpose was to identify the thermo-optical characteristics of layers with different refractive indexes. Thermo-optical studies were performed to determine the temperature hysteresis of optical parameters. These measurements showed that after annealing up to 300 °C and subsequent cooling, the value of optical parameters returned to the initial values.

## 1. Introduction

Amorphous SiN_x_:H layers have been known since the eighties and are still used for various optical applications. They are mainly used as very effective antireflection coatings in the production of crystalline silicon (c-Si) and multicrystalline silicon (mc-Si) solar cells [1,2]. The hydrogen content in the layers effectively passivates the material’s surface and suppresser surface recombination [3,4,5,6,7,8]. SiN_x_:H layers with various levels of nitrogen content are widely used in optoelectronics. Their main applications are as gate dielectrics in thin film transistors, in photonics, phototransistors and image sensors, or as multilayer systems for antireflection coatings in the infrared range [9,10,11,12]. The plasma-enhanced chemical vapour deposition (PECVD) technique is currently one of the most preferred deposition methods for the a-SiN_x_:H layers. This is due to the lower energy consumption during the process and the high hydrogen content in the layer [13,14,15]. These layers can also be obtained by other CVD techniques, such as atomic layer deposition (ALD) and catalytic (Cat-CVD), and other layer techniques, such as magnetron sputtering and ion-implantation [16,17,18,19]. It is known that the properties of a-SiN_x_:H layers, such as the extinction coefficient and the optical band gap, can be adjusted by varying the composition ratio of Si and N or by changing the deposition temperature [20,21]. There has been a lot of research on the effect of the processing parameters (Si/N ratios and Si-H/N-H bond ratios in layers deposited) on the optical parameters of silicon nitride films grown from a SiH_4_, NH_3_ and N_2_ mixture [22,23]. The authors present the relationship between the Si/N ratio and refractive index [24]. It is known that, as the nitrogen content of the layer decreases, the refractive index value increases [25].

In this paper, we report unpublished results of the optical and thermo-optical properties of the hydrogenated amorphous silicon nitride layers—specifically conventional silicon nitride (SiN_x_) and silicon-rich silicon nitride (Si-rich SiN_x_) deposited on Si(001) and on glass (fused quartz). We present their optical and thermo-optical analyses before and after a heating process to temperatures up to 300 °C. The layers were deposited using a wide gaseous flow range [SiH_4_]/[NH_3_], thus receiving a wide range of the Si/N ratio in the layers. The calculated thermo-optic coefficient (TOC) was used for the thermal characterisation of optical properties. Thermo-optical studies have indicated that heating and cooling cycles are repeatable; the values of *n* and *k* almost coincide with the values after heating/cooling processes. In the literature, thermos-optical results for SiN:H layers containing different contents of nitrogen have been presented, but not in the wide range we present. Our study allowed us to find a nonlinear dependence of the refractive index on the nitrogen content of the film.

## 2. Materials and Methods

### 2.1. Technology and Samples

For the purpose of this work, two series of samples were fabricated using the PECVD technique. All the layers were deposited on crystalline silicon (001)—oriented from gaseous silane and ammonia, and fused quartz (samples 6–8). Before deposition, the substrates were cleaned: samples 1–5 were preliminary washed in acetone and isopropyl alcohol and then placed in the reactor; for samples 6–8, organic matter was first removed from the wafer surfaces using ultrasonic cleaning in an acetone solution for 20 min then immersion in 5% HF acid to remove the native oxide layer before being transported into the deposition chamber.

The deposition was performed by the PECVD method using the Plasmalab System 100 from Oxford Plasma Technology and the multi-module MW-RFCVD Elettrorava system. The excitation frequencies were a low frequency, (LF) 90–450 kHz, and a high frequency (HF), 13.56 MHz. The layers were obtained at 220 °C and 310 °C from silane and ammonia, used as the gas mixture. The chamber pressure was kept constant (53 Pa) during the obtained process. Layers 6 and 7 were obtained at HF and LF frequencies alternately during one deposition, and other layers only at the HF frequency. Directly before layer deposition, the substrates were etched for 10 min of cleaning in an argon plasma environment. The other process parameters are presented in Table 1.

### 2.2. Characterization Methods

The measurements of the chemical composition and surface morphology for the obtained layers were performed on a scanning electron microscope (NOVA _NANO_SEM 200, FEI EUROPE Company, (Oregon, OR, USA) and an energy dispersive spectrometer (EDAX Genesis, Paoli, PA, USA). The chemical composition and Si/N ratio in the layer were evaluated by the EDX technique (Genesis, Paoli, PA USA). Obtained films were also characterised by X-ray diffraction (XRD, X’Pert Philips, Almelo, Netherlands) using the Philips X’Pert diffractometer with a CuKα radiation source (λ = 1.54056 Å) for 2θ = 6–90° with a step size of 0.01°. The layers were analysed with FTIR (Brucker, Billerica, MA, USA) and spectroscopic ellipsometry (SE, manufacturer.A.Woollam, Lincoln, NE, USA). Subsequently, the Fourier transform infrared (FTIR) spectrum of layers was measured within the range of 400–4000 cm^−1^ using the Vertex Brucker 70 V (Brucker, Billerica, MA, USA) (resolution 4 cm^−1^, 275 scans). Atomic force microscopy (AFM, XE-70 Park System Corp., Suwon, Korea) imaging was performed using a CoreAFM scanning-probe microscope (Nanosurf AG, Liestal, Switzerland) operating in a contact-force mode. A 256 × 256 pixel resolution was used for all AFM topography images.

The thermo-optic properties of SiN layers have been examined using the spectroscopic ellipsometry (SE) technique. SE measurements were performed with the use of a Woollam M-2000 ellipsometer (Woollam Co. Inc., Lincoln, NE, USA) with a spectral range of 193 to 1690 nm. The M-2000 ellipsometer was additionally equipped with a heat cell to change the sample temperature. Samples were heated from room temperature up to 300 °C in an air atmosphere under normal pressure conditions.

SE is a nondestructive and noncontact method that enables measuring such optical properties of thin films as optical constants, thickness and energy bandgap. The changes in a polarisation state have been registered as ellipsometric Ψ and Δ spectra for an incident angle of 70°. Ellipsometric parameter Ψ represents the amplitude ratio and the Δ parameter represents the phase difference between *p* and *s* polarised light waves. Both parameters Ψ and Δ are related and fulfil the fundamental equation of ellipsometry (Equation (1)) [26,27]:(1)$$p\;=\;\tan\Psi \cdot e^{i\Delta }$$

In order to determine dispersion relations of optical indices, namely refractive index *n*(*E*) and extinction coefficient *k*(*E*), the Tauc–Lorentz (TL) parameterisation model [28,29] has been applied. The Tauc–Lorentz model is Kramers–Krönig (K–K) which is consistent and physically correct; therefore, it is successfully employed for amorphous semiconductors and dielectrics [30,31,32]. The imaginary part of the dielectric function *ε*_2_ in the TL model is a result of the multiplication of the Lorentz oscillator function *ε_L_*(*E*) (Equation (2)) and the Tauc function *ε_T_*(*E*) (Equation (3)):(2)$$\varepsilon_{L}\left(E\right)\;=\;\frac{A_{L}\cdot E_{g}\cdot \Gamma \cdot E}{{\left(E^{2}-E_{0}^{2}\right)}^{2}+\Gamma^{2}\cdot E^{2}}$$
(3)$$\varepsilon_{T}\left(E\;>\;E_{g}\right)\;=\;A_{T}{\left(\frac{E-E_{g}}{E}\right)}^{2}$$
where *A_L_* is the strength of the imaginary part of the *ε*_2_ dielectric function, Γ is the broadening of the peak and *E*_0_ is the central energy of the peak, *A_T_* is the Tauc coefficient, *E* is the photon energy and *E_g_* is the optical bandgap. The *ε_T_*(*E*) function is equal to zero below the bandgap.

In order to obtain the best agreement between the model and the measurement data, the Gauss model was additionally used in optical modelling. The Gauss oscillators are often used with the Tauc–Lorentz model as multiplicative factors. The imaginary part of the dielectric function of the Gauss model *ε_G_*(*E*) is given in Equation (4) [33]:(4)$$\varepsilon_{G}(E)\;=\;A_{G}\left(\exp\left[-{\left(\frac{E-E_{0}}{\sigma }\right)}^{2}\right]+\exp\left[-{\left(\frac{E+E_{0}}{\sigma }\right)}^{2}\right]\right)$$
where
$$\sigma \;=\;\Gamma /2\sqrt{ln2}$$
and *A_G_* is the amplitude.

The imaginary and real part of the dielectric function are linked by the K–K relations. Therefore, the real part of the dielectric function of the Tauc–Lorentz and Gauss model can be calculated based on K–K consistency.

Thermo-optical properties of transparent films have been described using the Prod’homme theory [34]. The most important parameters describing optical and physical properties are the thermo-optical coefficient (TOC) and the thermal expansion coefficient (TEC). The TOC is defined as a derivative of the refractive index relative to temperature TOC = *dn*/*dT*, while TEC is defined as the thickness derivative over temperature
$$\mathrm{TEC}=\frac{d(thickness)}{dT}.$$


## 3. Results

### 3.1. Morphology and Chemical Structure of Layers

SEM and AFM morphology studies performed of all SiN samples confirmed the high quality of the produced amorphous layers (shown in Figure A1 in Appendix A section). All samples were very similar in terms of surface morphology. The surface exhibits high homogeneity without any large-scale defects, cracks. It is smooth and uniform. Only in a small area, few impurities are visible (Figure A1a and Figure A2 in Appendix A section). The surface roughness (RMS value), determined on the basis of AFM tests for all tested samples, was very similar to each other and was approximately 7 nm.

The chemical composition of the layers was roughly estimated by the EDS technique (Genesis, Paoli, PA, USA) and the value of the Si/N ratio is presented in Table 2. The amorphous of the SiN layers was confirmed by X-ray diffraction measurements (X’Pert Philips, Almelo, The Netherlands). In all tested samples, there were diffraction reflections corresponding to Si (001) substrate and a small amount of impurities (see Appendix A section Figure A3).

The structural characterisation of the a-SiN:H layers were performed by means of FTIR spectroscopy in the absorption mode, and in the range of 400 to 4000 cm^−1^.The FTIR absorption spectra are typical for a-SiN:H layers containing the peaks of chemical bonds Si–N, N–H, Si–H and were presented in papers [11,35]. The assignments of the infrared vibration modes of the a-SiN:H layer are reported in Table 3.

It was observed that with the increase of the gas-flow ratio, the intensity of bond peaks of Si–N (820–890 cm^−1^), Si–H, and N–H (at 1139 cm^−1^) increases, and the intensity of the bond peaks of N–H (at 3343 cm^−1^) decreases. With the increased s ratio, more connections with nitrogen are created (N–H bending at 1139 cm^−1^). In the case of the N–H stretching bond (at about 3343 cm^−1^), its density decreased with s.

### 3.2. Optical Studies

All ellipsometric studies were performed for three samples on silicon SiN6, SiN7, and SiN8 for a 70° angle of incidence across the full spectral range (193 to 1690 nm or 0.7 to 6.5 eV). The same three SiN layers have been applied on fused silica, which enabled us to measure transmission spectra. The data were analysed using CompleteEASE 5.15 software. Spectral dependences of Ψ and Δ for three samples are presented in Figure 1a–c. The mean squared error (MSE) values determined using the Levenberg–Marquardt method are presented in Table 4. The Tauc–Lorentz model was used to approximate the experimental results. The value of MSE confirms the fact that experimental data are in good agreement with the applied TL model. Small discrepancies between the measurement data and the applied model are visible in the UV region. The following parameters from the TL model at various wavelengths are summarised in Table 4: bandgap *E_g_*, amplitude A, broadening B, and refractive index values. The *E_g_* values strongly depend on the layer preparation process conditions. The determined values of *E_g_* have been confirmed in other works [38,39].

On the basis of the applied model, the dispersion relations of the refractive index *n*(*E*) and extinction coefficient *k*(*E*) for three SiN layers on silicon are shown in Figure 2. As can be observed, hydrogenated amorphous silicon-rich nitride exhibits high transparency in the photon energy range of 0.7 to 2.2 eV, i.e., in the visible and near-infrared spectrum area.

Figure 3 shows the refractive index *n* value at 632 nm as a function of the Si/N ratio for the samples presented in Table 2. As can be seen, a strong dependence of refractive index versus Si/N ratio (above 7) occurs. For the low Si/N ratio, it is difficult to determine the relationship between the refractive index versus Si/N.

In Figure 4, the influence of temperature on the thickness of SiN layers (SiN6, SiN7, and SiN8) in the heating and cooling cycle is shown. The difference of thickness before and after temperature treatment is about 2 nm, which means that all SiN layers are thermally stable. For the temperature range of 25 to 300 °C, the TEC values were determined at 400 nm and are presented in Table 5.

The thermal hysteresis of refractive index *n* for various wavelengths (400, 632, 900, and 1500 nm) is presented in Figure 5. The changes of *n* with the temperature are linear, i.e., there is no form of a hysteresis loop. Refractive index values at 25 °C before and after temperature treatment are nearly identical. This enabled us to draw the conclusion that this thermodynamical process is reversible. Thus, the tested layers can be successfully applied in optics. The exception is *n*(*T*) at 1500 nm for SiN6, where the difference between the initial and final values of *n* is about 0.01. For the temperature range of 25 to 300 °C, the TOC values were determined at various wavelengths and are presented in Table 5.

The refractive index of the amorphous layers is a linear function of temperature and can take positive and negative values. The slope *dn*/*dT* (TOC) is given by differentiating the Lorentz–Lorenz equation, which is a function of the refractive index, polarisation coefficient *Φ* and volumetric expansion coefficient *β*, which is given by express as Equation (5):(5)$$dn/dT\;=\;\frac{\left(n^{2}-1\right)\left(n^{2}+2\right)}{6n}\;\cdot \;\left(\Phi -\beta \right)$$

If *β* > *Φ* then TOC becomes negative. As can be seen from the formula, the slope may change as a result of the refractive index *n* dispersion. Therefore, the slopes *dn*/*dT* for different wavelengths may change (for zero dispersion, the slopes should be the same). Figure 6 shows the thermal dependences of extinction coefficient *k*(*T*) at 400 nm. The linear dependence of the extinction coefficient *k* versus temperature appears for energy above the absorption band (2.4 eV). For the SiN8 sample, the extinction coefficient slightly changes with temperature, but the changes of *k* for SiN6 and SiN7 samples are negligible. Sample SiN8 exhibits a higher extinction than other tested samples. Such high *k* values affect the positive sign of the TOC coefficient in the absorption area.

The optical parameters of the tested layers are presented in Table 4 and TOC, TEC and thickness values are presented in Table 5. The TOC depends mainly on the polarizability of the layer and on the value of the TEC. If the TEC value is high, the TOC takes a negative value. The TOC values determined for 400 nm and 500 nm wavelengths are positive for the SiN8 sample. This is due to the absorption in the SiN8 sample, for which the temperature dependence is shown in Figure 7. The TEC values presented in Table 5 are high compared to relating to other materials [40,41,42].

We observed that the TOC values for the SiN6, SiN7 and SiN8 samples depend on the wavelength, while the TOC values for the SiN layers, presented in other works [35,43], do not show such a relationship. The slope of the *n*(*T*) dependence is nearly the same for all wavelengths in the transparency region. In this work, we observed a strong dependence of TOC values versus wavelength. Additionally, the inclination angle of linear dependence *n*(*T*) for a particular sample decreases with the wavelength, and thus the TOC value decreases. For samples presented in the literature [35,43], the value of the refractive index *n* increased with temperature. For SiN layers presented in this work, the *n* values decreased with increasing temperature.

These results are a consequence of the much higher gas-flow ratios for samples 2–5 deposited at only HF frequency. The second reason is the different synthesis methods used in samples preparations. Two frequencies, LF and HF, were used in the process of producing the SiN6, SiN7, and SiN8 samples.

The transmission spectra presented in Figure 8 were measured for the layers applied on the fused silica and were included during the fitting of the TL model to the ellipsometric results.

## 4. Conclusions

This paper presents the optical and thermo-optical dependences for non-stoichiometric silicon nitride layers obtained by two independent PECVD methods with varying levels of nitrogen content. The presented SiN layers were fully amorphous, without large-scale defects and cracks. The ellipsometric investigations showed that:
Refractive indices of the SiN layers in a wide range from 1.8 to 3.7 strongly depend on the volumetric composition of Si and N components. Measurement results for SiN layers in such a wide range have not been published in the scientific literature so far.The energy gaps of tested SiN layers determined from optical measurements range from 2.22 to 2.25 eV.

From thermal ellipsometric measurements performed for the temperature range of 20 to 300 °C, the spectral dependence and the temperature hysteresis of optical constants were determined. They proved that:
Thermo-optical parameters of the SiN layers strongly depended on the technological process.The values of the TOC parameter obtained for the samples prepared with a double frequency of radiation were negative, while the thermo-optical coefficient was positive for SiN layers deposited for one frequency of PACVD radiation.Thermo-optical measurements showed that after annealing up to 300 C and subsequent cooling, the value of optical parameters of layers returned to the initial values, which indicate the reversibility of their thermal properties and thermal stability.

## Figures and Tables

**Figure 1 materials-15-02260-f001:**
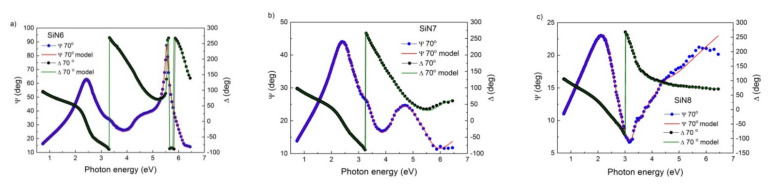
Spectral dependence of ellipsometric angles Ψ and Δ for (**a**) SiN6, (**b**) SiN7, and (**c**) SiN8 samples.

**Figure 2 materials-15-02260-f002:**
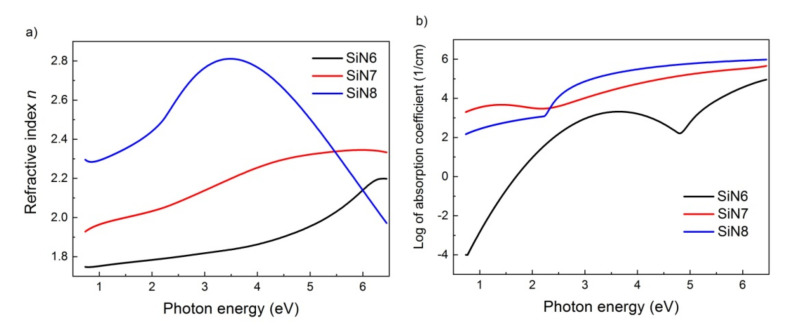
Dispersion relations of (**a**) refractive index *n* and (**b**) extinction coefficient *k* for SiN6, SiN7, and SiN8 samples.

**Figure 3 materials-15-02260-f003:**
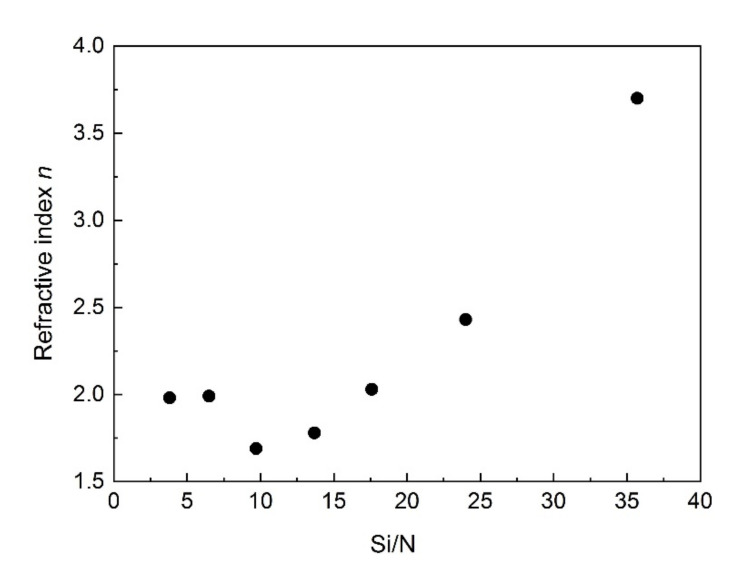
Refractive index as a function of Si/N ratio.

**Figure 4 materials-15-02260-f004:**
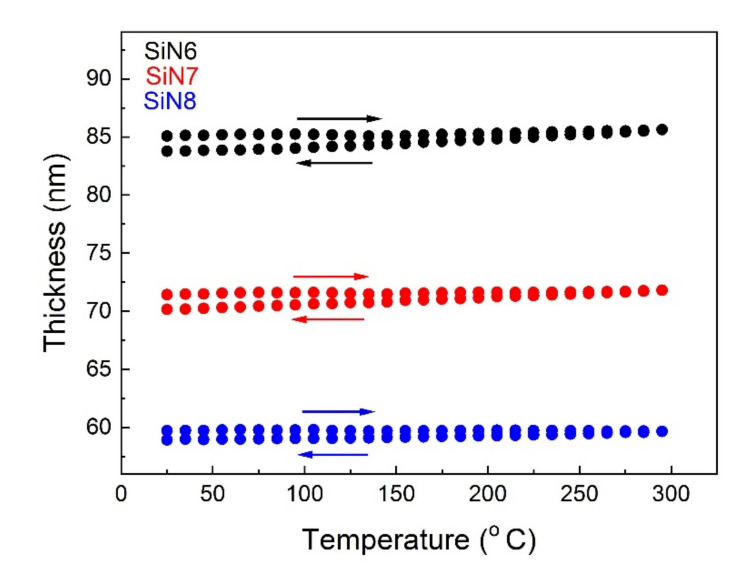
Thermal hysteresis of thickness for SiN6, SiN7, and SiN8 samples at 400 nm.

**Figure 5 materials-15-02260-f005:**
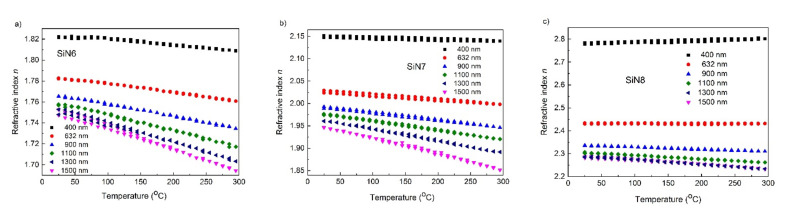
Thermal hysteresis of refractive index *n* for (**a**) SiN6, (**b**) SiN7, and (**c**) SiN8 samples.

**Figure 6 materials-15-02260-f006:**
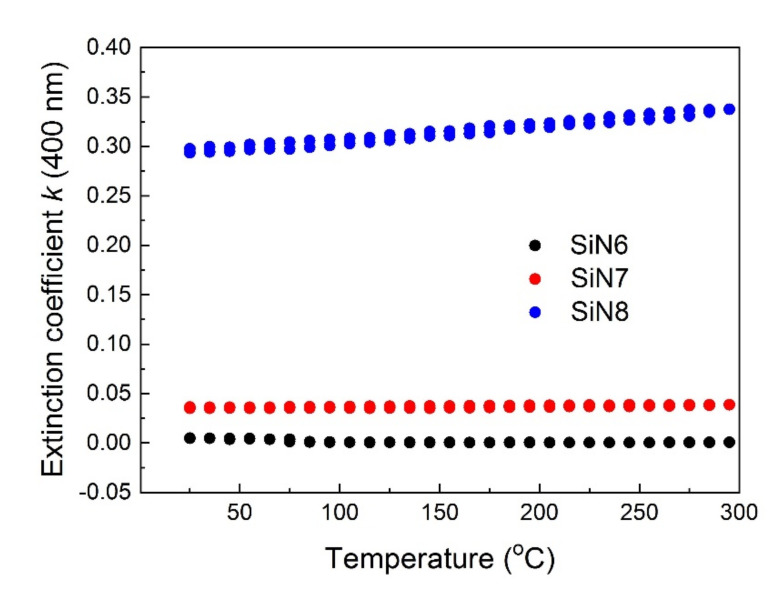
Thermal hysteresis of extinction coefficient *k* at 400 nm for SiN6, SiN7, and SiN8 samples.

**Figure 7 materials-15-02260-f007:**
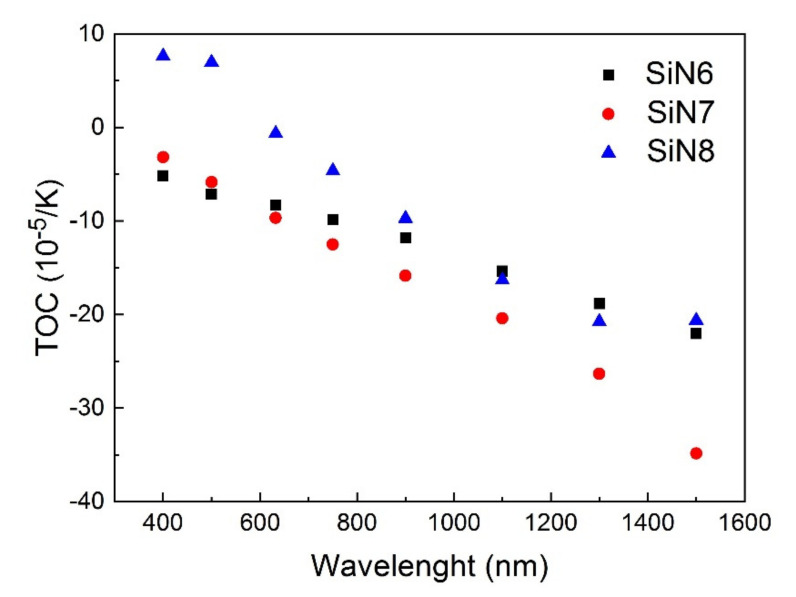
Dependence of thermo-optical coefficient TOC versus wavelength.

**Figure 8 materials-15-02260-f008:**
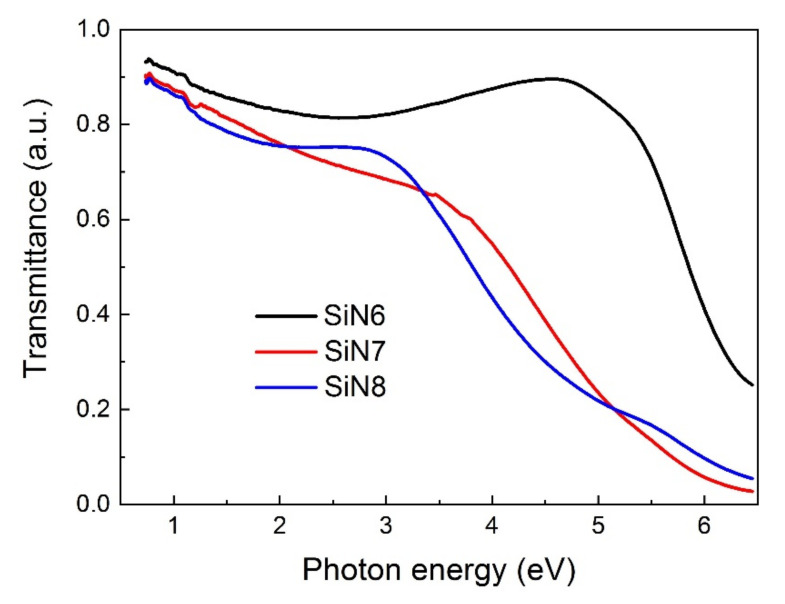
Spectral dependencies of the transmission of SiN6, SiN7, and SiN8 on fused silica.

**Table 1 materials-15-02260-t001:** The technological conditions of a-SiNx:H layers deposited in PECVD system.

Technique	PECVD System								
Series of Samples	Gaseous Substrate Flow [sccm]		s = [SiH_4_]/[NH_3_]	P HF [W]	Cycletime [s]	P LF [W]	Cycle Time [s]	Substrate Temperature [C]	Total Time [s]
	NH_3_	SiH_4_							
1	29	4	0.13	4	1800	-	-	220	1800
2–5	20–60	40–80	0.25; 0.42; 0.6; 1;	50	480	-	-	220	480
6–8	95, 20 *, 10 **	50, 460 *, 800 **	0.5; 23 *; 80 **	20	40 40 * 144 **	20 20 * - **	8 8 * - **	310	310 220 * 158 **

* value for sample 7. ** value for sample 8.

**Table 2 materials-15-02260-t002:** The value of ratio Si/N for all samples from EDS technique.

Samples	SiN1	SiN2	SiN3	SiN4	SiN5	SiN6	SiN7	SiN8	SiN9
[Si]/[N]	2.28	3.50	5.50	6.50	9.70	13.68	17.58	24.00	35.70

**Table 3 materials-15-02260-t003:** Wavenumbers for the principal atom bondings in amorphous silicon nitride [36,37].

Wavenumber (cm^−1^)	Bonding	Assignment
820–890 1139 2163 3343	Si–N N–H Si–H N–H	Si–N stretching N–H bending SiH_2_ stretching in N_3_Si–H N–H in band Si_2_N–H

**Table 4 materials-15-02260-t004:** Optical parameters and MSE values of SiN6, SiN7, and SiN8 samples.

Sample	MSE	A	B	E_g_ [eV]	n _400 nm_	n _632 nm_	n _900 nm_	n _1500 nm_
SiN6	5.8	1.16	0.71	2.23	1.82	1.78	1.77	1.75
SiN7	10.7	6.16	0.38	2.25	2.15	2.03	1.99	1.95
SiN8	7.5	131.39	7.22	2.22	2.78	2.43	2.33	2.28

**Table 5 materials-15-02260-t005:** Thermo-optical coefficient, thermal expansion coefficient and thickness values for SiN6, SiN7, and SiN8 samples.

Sample	TOC [10^−5^/K]								Thickness [nm]	TEC [10^−3^ nm/K]
	λ									
	400 [nm]	500 [nm]	632 [nm]	750 [nm]	900 [nm]	1100 [nm]	1300 [nm]	1500 [nm]		
SiN6	−5.16	−7.16	−8.28	−9.87	−11.80	−15.34	−18.84	−22.03	85.07	7.04
SiN7	−3.17	−5.84	−9.66	−12.50	−15.86	−20.40	−26.33	−34.83	71.42	6.12
SiN8	7.61	6.95	−0.64	−4.63	−9.72	−16.30	−20.75	−20.61	59.73	6.12

## Data Availability

Not applicable.

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
