# Peer review of "The Optical and Thermo-Optical Properties of Non-Stoichiometric Silicon Nitride Layers Obtained by the PECVD Method with Varying Levels of Nitrogen Content"

_materials, 2022, doi:10.3390/ma15062260_

Round 1
Reviewer 1 Report
The article is not sound.
Author Response
Dear Reviewer
in accordance with your comments we have made changes to the text of the article, made corrections, improved the quality of figures and models, and edited the summary.
Reviewer 2 Report
This paper highlights the optical and thermo-optical properties of a-SiNx: H layers obtained using the PECVD technique. The paper cannot be accepted in the present form as it needs further improvements.
-I cannot see what is new in this paper versus prior work. I encourage the authors to pick those papers closest to this work and explain the key differences and contributions. I advise the authors to avoid using ambiguous language in explaining their novelty because it makes the technical assessment challenging, negatively impacting the outcome.
- The problem reported has already been covered by a series of articles published. I struggle to see any novelty significant new contribution.
-I would encourage the authors to present a more compelling argument regarding the originality and relevance of their work relative to the previous research landscape in this area in a clear, easy-to-understand, and verifiable fashion.
- Abstract: The text must be carefully revised. Some sentences contain mistakes. Avoid using the acronym. Explain it in detail. Discuss what the Research Gaps/Contributions are?
- The English writing of the paper is required to be revisited. Please check the manuscript carefully for typos and grammatical errors. Avoiding split infinitives can help your writing sound more formal.
- Line No: 16 - It seems that there is an article usage problem here. The verb did not seem to agree with the subject. Consider changing the verb form.
- Line No: 20 - The singular noun purpose follows a number other than one. Consider changing the noun to the plural form.
- Line No: 21- It appears that the singular demonstrative This is modifying the plural noun measurements. Consider using a plural demonstrative or a singular noun instead.
- Line No: 30, 32-33, 38, 47, 81-85, 96-98, 126, 137-141, 147-151, 189, 195-199, 205, 214-216, 233, 243-245, 250-254, 263-270: Your sentence may be unclear or hard to follow in many parts. Consider rephrasing.
- Equation 5 need to be typed instead of cropping and pasting it.
- All figures are of poor quality; try to replace them with good resolution.
-Provide a proper reference for the equations. It is well known and available in much literature. You can consider removing a few.
-Discussion Section: Introduce a new section, "Discussion," with more current references, which compare the results obtained by the authors with other studies carried out by other researchers. Conclusions should be more concrete. They should be summarized in 3-5 bullet points that clearly show the conclusions of this study. In addition, since it is a review, it is essential to indicate future lines of research.
- The manuscript needs to be checked twice before submitting. It has many flaws, and the track changes/different font colors/formatting issues need to be rectified beforehand.
The list could go on, but the bottom line is that the authors need to rewrite the paper or reconsider the research content before being considered for publication in this journal.
Author Response
Dear Reviewer
in accordance with your comments we have made changes to the text of the article, made corrections, improved the quality of figures and models, and edited the summary.
Responses to individual comments:
- -I cannot see what is new in this paper versus prior work. I encourage the authors to pick those papers closest to this work and explain the key differences and contributions. I advise the authors to avoid using ambiguous language in explaining their novelty because it makes the technical assessment challenging, negatively impacting the outcome.
- - The problem reported has already been covered by a series of articles published. I struggle to see any novelty significant new contribution.
- I would encourage the authors to present a more compelling argument regarding the originality and relevance of their work relative to the previous research landscape in this area in a clear, easy-to-understand, and verifiable fashion.
Response:
The presented work is based on the results of independent studies of a-SiN samples with the widest possible spectrum of nitrogen content.
Admittedly, there have been presented in the literature thermo-optical results for layers containing different contents of nitrogen, but not in such a wide range as we present. Our study allowed us to find nonlinear dependence of the refractive index on the nitrogen content of the film. This observation has been introduced in the Introduction section.
- Abstract: The text must be carefully revised. Some sentences contain mistakes. Avoid using the acronym. Explain it in detail. Discuss what the Research Gaps/Contributions are?
Response: The text was revised and was changed. The acronyms have been expanded.
- The English writing of the paper is required to be revisited. Please check the manuscript carefully for typos and grammatical errors. Avoiding split infinitives can help your writing sound more formal.
- Line No: 16 - It seems that there is an article usage problem here. The verb did not seem to agree with the subject. Consider changing the verb form.
- Line No: 20 - The singular noun purpose follows a number other than one. Consider changing the noun to the plural form.
- Line No: 21- It appears that the singular demonstrative This is modifying the plural noun measurements. Consider using a plural demonstrative or a singular noun instead.
- Line No: 30, 32-33, 38, 47, 81-85, 96-98, 126, 137-141, 147-151, 189, 195-199, 205, 214-216, 233, 243-245, 250-254, 263-270: Your sentence may be unclear or hard to follow in many parts. Consider rephrasing.
Response: The English writing of this paper was revised and changed.
- Equation 5 need to be typed instead of cropping and pasting it.
Response: Equation 5 has been typed as suggested.
- All figures are of poor quality; try to replace them with good resolution.
Response: All figures have been corrected and resolution has been increased.
- -Discussion Section: Introduce a new section, "Discussion," with more current references, which compare the results obtained by the authors with other studies carried out by other researchers. Conclusions should be more concrete. They should be summarized in 3-5 bullet points that clearly show the conclusions of this study. In addition, since it is a review, it is essential to indicate future lines of research.
Response: Thank you for reviewer suggestion, Conclusions section was extender.
The most important results achieved in the work have been given in points.
Reviewer 3 Report
This paper reports a study on the optical properties as well as the effects of thermal treatments on the optical properties of SiNx thin films. In general, the study contains interesting new progress, which is worth of being published. I suggest the authors make revision according to the suggestions below:
- The abstract part, there is only one sentence for the major results. I think this is far from enough for readers to obtain major information from the abstract. Furthermore, baseline seems to be not precise for the description here.
- Porosity is different from surface roughness. I suggest the authors use RMS for reporting the surface roughness. Most amorphous films have point defects. I suggest the authors use no large-scale defect was observed in the film instead of defect.
- The refractive index data in Figure 3 seems confusing to me. Does it mean the less Si the lower RI? the Si/N ratio can not reach very low practically. I found the data at Si/N ~ 5 the RI was quite close to traditional SiNx. I suggest the authors revise the expression for better understanding.
- What is the impurity? why it has only one peak at higher angle in XRD pattern?
Author Response
Dear Reviewer,
in accordance with your comments, we have made changes to the text of the article, made corrections, improved the quality of figures and designs, and edited the summary.
Responses to individual comments
- The abstract part, there is only one sentence for the major results. I think this is far from enough for readers to obtain major information from the abstract. Furthermore, baseline seems to be not precise for the description here.
Response: According to the reviewer's comment, the most important results achieved in the work have been added in abstract part.
- Porosity is different from surface roughness. I suggest the authors use RMS for reporting the surface roughness. Most amorphous films have point defects. I suggest the authors use no large-scale defect was observed in the film instead of defect.
Response: Thank you for reviewer suggestion, the sentence has been changed in text.
- The refractive index data in Figure 3 seems confusing to me. Does it mean the less Si the lower RI? the Si/N ratio can not reach very low practically. I found the data at Si/N ~ 5 the RI was quite close to traditional SiNx. I suggest the authors revise the expression for better understanding.
Response: The general trend is that the refractive index increases with increasing Si in the silicon nitride layer. However, for high nitrogen contents in the layer, the refractive index may also depend on the technological deposition conditions. For higher nitrogen contents in the SiN:H layer, the refractive index takes values in the range of 1.8-2.2, which are typical for this type of layers. The authors of papers 35 and 43 prove that at low nitrogen contents in silicon nitride layers the coatings are rich in silicon and their refractive index is high above 3.5. Such coatings have the characteristics of thin films, (interference occurs in them)
- What is the impurity? why it has only one peak at higher angle in XRD pattern?
Response: It is a calcium carbonate that was accidentally found on the surface. This has been confirmed by complementary SEM, EDS and XRD studies performed after basic studies including ellipsometry. Baseline studies were performed on a clean surface. The diffraction reflectances are shown on a logarithmic scale. The amount of impurities is at the detection limit of the XRD method, so other remaining diffraction maxima are not visible.
Round 2
Reviewer 2 Report
I agree with the comments made and accept the changes made. I consider the text worthy of publication.
This manuscript is a resubmission of an earlier submission. The following is a list of the peer review reports and author responses from that submission.
Round 1
Reviewer 1 Report
The paper represents a collection of different data representing amorphous SiNx:H layers.
However, the motivation and systematicity are missing.
- Abstract and Introduction. Why namely these samples? What problems were intended to be addressed and solved? What is the main message to the reader? What are the main scientifically sound achievements, except of the new experimental data?
- Table 1. Why namely these parameters were used? Again, what is systematic approach?
- P. 4. What does it mean “exemplary sample”?
- Fig. 1. I do not see anything in Figs 1a and 1b.
- Fig. and upwards, Tables 4 and upwards – why only samples 6 to 8 are compared?
- Conclusions. I did not find ANY conclusions just the summarized results. Again, what are the new findings, and the superiority of the results as compared to the previous published numerous investigations.
The general impression is that the manuscript is prepared on the basis of the MSc (or probably PhD) thesis. However, it is not suitable for the publication in the journal, as it is not interesting to the wider audience.
Author Response
Dear reviewer,
our responses to the reviewer's comments and suggestions are included in the attached file.

Reviewer 2 Report
This article presents the results of the formation by the PECVD method of films of hydrogenated amorphous silicon nitride with different compositions (stoichiometric and silicon-enriched) on silicon and fused quartz substrates. For them, the structure and morphology of the surface, as well as the optical properties were investigated by far IR spectroscopy, spectroscopic ellipsometry, and transmission spectroscopy on fused silica substrates. The main task of the authors was to study the thermo-optical characteristics of the formed films and their thermal stability when heated to 300 ° C. In situ optical studies were carried out using a special optical attachment inserted into the optical compartment of the spectral ellipsometer. To analyze the ellipsometric data, we used standard optical models (Tauc-Lonentz, Lorent oscillator function, Tauc function, and Gauss model) with subsequent reconstruction of the real part of the dielectric function within the framework of the Kramers-Kronig integral relations. Thermo-optical studies and simulation of experimental data showed that repeated heating and cooling cycles do not lead to noticeable changes in the refractive index and extinction, which confirmed the preservation of the stability of the films under study in the case of their composition close to stoichiometric. And for films of hydrogenated amorphous silicon nitride enriched with silicon, noticeable changes in the refractive index are observed in the region of intrinsic absorption.
The presented data and their model calculations are not supported by an analysis of the energy band structure. The results obtained are new and may be of interest to researchers in the field of hydrogenated amorphous silicon nitride.
Notes on the manuscript of the article:
- SEM and AFM images in Fig. 1 (a, b) are uninformative because they look exactly the same. At such low magnifications (1000x and 10000x) and RMS roughness, there is no need to demonstrate them. In Fig. 1 s, too large a format of 10x10 μm2 was chosen, which does not allow us to examine the details of the morphology of the samples. You need a format 1x1 μm2 or less in the x and y coordinates, and in the z coordinate - keep the scale of 10 nm, but correct the background subtraction. The slope is several microns over a length of 10 microns.
- The Appendix (Fig. A1) shows the CRD spectra with an incorrectly selected scale along the y-axis. Due to the fact that the silicon substrate gives too large a signal, the gallo from the amorphous phase, which the authors constantly talk about, is not visible on a linear scale. The figure should be presented on a semi-log scale along the y-axis. In this case, with an increase in the nitrogen content in the a-SiNx-H film, the shape and position of the gallo maximum should change.
- When discussing these spectra in the far IR region of the spectrum, the authors do not give the spectra themselves, but only state an increase in the amplitudes of the peaks of the chemical bonds Si-H (2163 cm-1) and NH (1139 cm-1) with an increase in the [Si] / [N] and, accordingly, a decrease in the amplitude of the NH chemical bond (3343 cm-1). At the same time, they do not explain the reasons for the change in the nature of the bond between nitrogen and hydrogen with an increase in the silicon content in the amorphous hydrogenated silicon nitride film.
- Why were the optical properties considered only for the samples of a-SiNx-H films obtained by the hybrid method at high and low frequencies sequentially? And the films obtained only by the high-frequency PESVD method have not been studied? What is the reason for this? Can we assume a different nature of the optical parameters and their changes with temperature?
- In the sentence: "The mean squared error (MSE) values ​​determined using the Levenberg-Marquardt method are presented in Table 1" there is an incorrect reference to the Table. Refer to Table 4.
- In the spectrum of the extinction coefficient for SiN6, SiN7, SiN8 samples, for a confident statement about the approximate conservation of the band gap, it is necessary to use a semi-log scale, due to the strong change in the extinction coefficient of about 2.2 eV for these samples. The authors argue that high transparency in the films is observed below 2.2 eV, but on a linear scale along the y axis it is impossible to understand the exact changes in the extinction coefficient for the selected three films with different compositions in terms of silicon concentration.
- In Fig. 4 does not indicate at what photon energies the values ​​of the refractive index were taken depending on the [Si] / [N] ratio. At the same time, the authors do not explain the reasons for the sharp increase in the refractive index at the ratio [Si] / [N]> 25.
- When discussing the dependence of the thickness on the cycle of heating to 300 ° C - cooling to room temperature, the authors claim that their thickness changes by tens of nm. Although from Fig. 5 it can be seen that these changes are limited to values ​​in units of nm. The reason for these changes in thickness is not discussed. Could this be related to a change in the morphology of the film? For example, with a decrease in roughness during annealing?
- In Fig. 6 shows the changes in the refractive index on the annealing temperature at different wavelengths. The graphs show poorly changes during cooling at wavelengths of 400, 632 and 900 nm. The changes are noticeable only for Fig. 6 (a) at 1500 nm. The authors do not attempt to explain the mechanism of such differences in the temperature dependences of the refractive index as a function of both composition and wavelength.
- When analyzing the temperature hysteresis of the extinction coefficient in the region of interband transitions (400 nm or 3.1 eV) (Fig. 7 and Table 5), the authors associate the sign and value of the temperature optical coefficient with the polarizability of the a-SiNx-H layer in the SiN8 sample, but not with a change in the probability of optical transitions with increasing temperature due to changes in the band structure and the appearance, for example, of a higher density of reduced states due to an increase in the silicon concentration. Changes in the energy band structure are the basis for changes in optical properties. These arguments from the words of the authors are absent and the cited literature [35, 43].
- In Table 1 and Table 5, the same letter “s” denotes two different parameters: flux ratio s = [SiH4] / [NH3] and s - film thickness.
- In the transmission region, the values ​​and sign of the temperature optical coefficient (Fig. 9) should be determined by the density of states in the band gap of the alloys under consideration and their change with increasing (decreasing) temperature.
- In the transmission spectra measured for SiN6, SiN7 and SiN8 samples on fused silica substrates, high transparency (about 40%) remains up to 4-6 eV, which contradicts the data on the band gap of 2.2 eV and the calculated spectra of the extinction coefficient (Fig. . 3 b).
The manuscript of the article requires a serious revision and an answer to the questions posed, as well as the elimination of noticed errors and typos.
The conclusions of the article are formulated in a very general way. Their specification is required.
Author Response

(The authors gave the same response as above.)

Round 2
Reviewer 1 Report
Practically no changes were made in the manuscript. It still appears as a scatter of random data obtained in a non-systematic way in different samples. Thus, I am once again recommending rejecting it.
Author Response
Dear Reviewer,
the content of our manuscript has been significantly changed compared to the original version. We believe that the experimental data presented in the manuscript are consciously selected in accordance with the assumptions of the work. These are not random data.
Reviewer 2 Report
The authors answered my main questions and gave detailed comments to them. They also made changes to the text of the manuscript and some of the figures, expanded the discussion of a number of results, supplemented the conclusion on the article, which undoubtedly improved the perception of the article materials.
There is a small note on Fig. 1A2, which shows the structure of the three investigated silicon nitride films. It is necessary to indicate what determines the contribution to the diffraction pattern from an amorphous silicon nitride film?
In Appendix A, it is necessary to introduce an additional interpretation of the diffraction peaks in Figure 1A2. Two of them belong to the peaks of the silicon substrate (33.5о and 69о), and the nature of the third peak at 62о needs to be clarified. The y-axis in all spectra must be scaled to get an idea of ​​the thickness of the silicon nitride film layer. In the original Figure 1A2, there was a noticeable difference in the intensities of the main peaks. And in the new version, this difference is not visible.
I believe that after these changes the manuscript can be recommended for publication.
Author Response
Dear Reviewer,
We fully agree with the reviewer's opinion on the interpretation of X-ray diffraction patterns. In the presented diffractograms there are visible three peaks. Two of them belong to the peaks of the silicon substrate (33.5о and 69о) and third peak corespond to small amout of crystalline impurities in the bulk of the films. Unfortunately, this reflex is insignificantly small and does not allow us to define the type of this phase. Figure 1 A2 has been supplemented with relevant explanations.